

# Earth's surface mass transport derived from GRACE, evaluated by GPS, ICESat, hydrological modeling and altimetry satellite orbits

Christian Gruber[1], Sergei Rudenko[2], Andreas Groh[3], Dimitrios Ampatzidis[4],

Elisa Fagiolini[1]

[1]*Helmholtz Centre Potsdam, German Research Centre for Geosciences (GFZ),*

*Section 1.2: Global Geomonitoring and Gravity Field,*

*c/o DLR Oberpfaffenhofen, Münchener Strasse 20, 82234 Wessling, Germany,*

*Tel.: +49-8153-90832-11, Fax.: +49-8153-90832-01, email: gruber@gfz-potsdam.de*

[2]*Deutsches Geodätisches Forschungsinstitut, Technische Universität München (DGFI-TUM),*

*Acrisstrasse 21, 80333 Munich, Germany, email: sergei.rudenko@tum.de*

[3]*Technische Universität Dresden, Institut für Planetare Geodäsie,*

*01062 Dresden, Germany, email: andreas.groh@tu-dresden.de*

[4]*Bundesamt für Kartographie und Geodäsie (BKG), Richard-Strauss-Allee 11,*

*60598 Frankfurt, Hessen, Germany, email: dimitrios.ampatzidis@bkg.bund.de*

## ABSTRACT

The Gravity Recovery and Climate Experiment (GRACE) delivers the most accurate quantification of global mass variations with monthly temporal resolution on large spatial scales. Future gravity missions will take advantage of improved measurement technologies such as enhanced orbit configurations and tracking systems as well as reduced temporal aliasing errors. In order to achieve the latter, sub-monthly to daily innovative models are computed. In addition, non-conventional methods





based on radial basis functions (RBF) and mascons will give the ability to compute models in regional and global representation as well. The present study compares for the first time a complete global series of solutions obtained by the RBF method with conventional solutions in order to quantify recent ice-mass changes. We further compare the ice-induced crustal deformations due to the dynamic loading of the crustal layer with the Global Positioning System (GPS) uplift measurements along Greenland's coastline. Available mass change estimates based on ICESat (Ice, Cloud, and land Elevation Satellite) laser altimetry measurements both in Greenland and Antarctica are used to asses the GRACE results.

A comparison of GRACE time series with hydrological modeling for various basin extensions reveals overall high correlation to surface and groundwater storage compartments. The forward computation of satellite orbits for altimetry satellites such as Envisat, Jason-1 and Jason-2 compares the performance of GRACE time variable gravity fields with models including time variability, such as EIGEN-6S4.

**Key words:**  GRACE – Radial basis functions – Kalman filtering – GPS deformation – Time-variable gravity field – Altimetry satellite orbits

## 1    INTRODUCTION

Since 2002 the Gravity Recovery And Climate Experiment (GRACE), Tapley et al. (2004) has been measuring temporal variations of Earth's gravitational field highly accurately. These data provide valuable information on the distribution and variation of mass in the Earth's sub-systems such as the atmosphere, hydrosphere, ocean and the cryosphere. The latest GRACE time-series of monthly gravity field solutions are computed in terms of spherical harmonic model coefficients at the German Research Centre for Geosciences (GFZ), version RL05a, University of Texas/Center for Space Research (CSR), version 05 and Technical University Graz, Institute of Geodesy (ITSG) version 2016. They show significantly less noise and spurious artifacts compared to their predecessors.

The Earth observation mission GRACE provides the only way to estimate groundwater storage changes on a global scale and in remote areas. Moreover, in order to gain further access to mass transport of short appearances, regional solutions in areas of strong anomalous signals need to be developed and new methods for their computation have to be investigated.



One candidate approach in this aspect is the transformation of the measurement data to in-situ (proxy) gravity observables with subsequent inversion and continuation by means of rigorous integral equations (Novák 2007). This non-conventional approach for the analysis of GRACE inter-satellite range observations, processed in combination with best knowledge reduced dynamic GRACE orbits has been elaborated in Gruber et al. (2014) and a detailed theoretical foundation of the method is presented in (Gruber et al., submitted to GJI). In brief, the transformed observations are first reduced by available geophysical background models and subsequently inverted as well as downward continued by a rigorous formulation in terms of reproducing kernel functions. Then, time-variable gravity field anomaly maps with respect to the subtracted background data have been derived. The observation equations are solved in spatial representation and are well suited for Kalman filtered solutions as covariance information is not required in spectral domain and can be applied to regional and insular domains only. This gives the opportunity to enhance the temporal resolution towards sub-monthly (weekly or daily) time series and to advance into local domains, thereby preserving the accuracy that is achieved from the standard monthly inversions.

A Kalman filter to derive daily gravity field solutions, first applied to GRACE data by (Kurtenbach et al. 2009; Kurtenbach et al. 2011), has been applied by us to the transformed GRACE gradient data. The main features are a stochastic process model for the data prediction step and the conversion of the range measurements to in-situ gravity observations. Standard integral equations are then used to solve for the gravity variations on surface grid tiles (Gruber et al. 2014). The applied Poisson kernel function thereby isotropically localizes the signal in spatial domain in contrast to a localization in spectral domain where global multi-pole moments (spherical harmonic coefficients) are estimated.

During least squares prediction, the surface grid tiles for the following day are recursively computed from the previous day and consecutively updated by the L1B observations in the Kalman gain. It should be noted that these solutions are constrained in two aspects. Firstly, by the applied background modeling that has been derived from available monthly GRACE solutions and trends as well as annual signal estimates thereof. Secondly, by the stochastic modeling of additional atmospheric and hydrological signal variations derived from geophysical models. It is therefore not necessary to post-filter the results as they do not exhibit GRACE-like anisotropic artifacts from the subsequent data inversion. Despite the regularized processing methodology, the system is well capable of capturing hydro-geophysical signals in their respective amplitudes.

First numerical results obtained using this method and their comparisons to standard





4    *Ch. Gruber et al.*

GRACE products were presented in (Dahle et al. 2016) and have been significantly enhanced since then. In this article we discuss the following evaluation methods with our latest results:

(i) Continental uplift rates from the Greenland GPS Network (G-NET) and Center for Orbit Determination in Europe (CODE)

(ii) Ice mass balances from ICESat

(iii) Hydrological basin comparison against the WaterGap hydrological model (WGHM)

(iv) Altimetry satellite orbits: Satellite Laser Ranging (SLR) and Doppler Orbitography and Radiopositioning Integrated by Satellite (DORIS) observation fits and arc overlaps

## 2   GREENLAND AND CONTINENTAL GPS-SITES COMPARISON

A significant spread of ice mass loss into northwest Greenland has been observed by GRACE and GPS during recent years (cf. Khan et al. 2010). We make use of monthly averaged vertical GPS site displacements from the Greenland GPS Network (G-NET), led by Ohio State University's division of Geodetic Science. G-NET is a network of 46 continuous GPS stations, installed on bedrock, spread across Greenland. We compare them with the crustal deformations inferred from post-filtered monthly GRACE gravity fields of ITSG2016 (Mayer-Gürr et al. 2016), GFZ Release 5a (Dahle et al. 2012), CSR Release 5 (Bettadpur et al. 2012) and the monthly averaged solutions derived from spherical radial basis functions (GFZ RBF). It should be noted that GPS site data are point values, whereas the GRACE solutions stem from area integrals. While this doesn't exclude direct comparison between the two data sets, insular discrepancies can be expected.

The simultaneous use of GNSS and GRACE data is a subject that has already been discussed in detail in the geodetic literature (e.g. Kusche and Schrama 2005, van Dam et al. 2007). The aforementioned publications focus on the comparison between the GPS and GRACE products, in terms of the regional or global mass distribution and/or the vertical displacements respectively.

We firstly complete all models with a center of mass to center of figure translation (degree 1, following Swenson et al. 2008). Changes in the ocean mass cause an offset between the center-of-mass and the center-of-figure frame, commonly denoted as geocenter motion. Briefly, any natural and anthropogenic water mass re-distribution at Earth's surface causes changes in global ocean mass. Net-inflow of fresh water and exchange between ice and water are typical phenomena that affect eu-static sea-level variability. The changes are reflected in the geocenter motion (degree 1) and are non-negligible for the GRACE mission. Since the





global eu-static sea-level variations are excluded from the de-aliasing model they can therefore

be derived empirically from the gravity field solutions.

Secondly, the Earth's flattening ($C_{20}$) being poorly observed by GRACE, has been replaced

by a satellite laser ranging (SLR) derived time-series from (Cheng et al. 2013) in the spherical

harmonic models (ITSG 2016, GFZ RL05a, CSR RL05). The flatting variations in the case

of the GFZ RBF solutions have remained unchanged after their co-estimation during Kalman

filtering.

The atmospheric and non-tidal ocean loading (GAC) is added back to the GRACE in-

ferred mass changes and the glacial isostatic adjustment (GIA) is removed from the temporal

GRACE coefficients using the GIA predictions according to the ICE-5G v1.3 model (Peltier

2004). This step is required to avoid propagation of gravity changes that are caused by the

vertical displacements from GIA into the lithosphere uplift calculation from GRACE, which is

obtained after a forward computation in the G-NET sites by means of viscoelastic load Love

numbers $k'_n$ and $h'_n$ according to Farrell (1972).

Finally, the named GIA-induced uplift from the ICE-5G v1.3 model is again restored,

thus the buoyancy effect at the base of the lithosphere (Wahr 1995) was taken into account.

In each site, the vertical displacements from the GPS time-series is then correlated with the

GRACE results (from monthly means) and computed over all stations.

Fig. 1 shows the correlations between the G-NET station uplift and the ice-induced crustal

deformations due to dynamic loading of the crustal layer obtained using the temporal gravity

field solutions: GFZ RBF and CSR RL05.

Main differences were found in the eastern part of Greenland, whereas only minor differ-

ences can be observed between the three spherical harmonic models (ITSG 2016, GFZ RL05a,

CSR RL05). The relatively lower correlations with G-NET around the eastern stations at

74°N, (DANE, HMBG, WTHG) can be explained by deficiencies in the GIA uplift model

(Dr. Ingo Sasgen, personal communication, July 6, 2017), that was therefore left out for the

computation of the average correlation numbers. These average correlations over the sta-

tions are very high, with some minor, insignificant deviations: GFZ RL05a: 90.2%, ITSG2016:

90.1%, CSR RL05: 89.6 % and GFZ RBF: 89.0% .

Then, the global GPS station network displacements from the Center of Orbit determina-

tion in Europe (CODE), computed by (Steigenberger et al. 2011) for the time span 2002-2012

have been treated accordingly. In Fig. 2, the correlations of the vertical station variations in-

ferred from GFZ GRACE RBF solutions and selected CODE GPS stations are displayed. Due

to minor differences between the individual solutions the GFZ RBF solutions are displayed



6    Ch. Gruber et al.

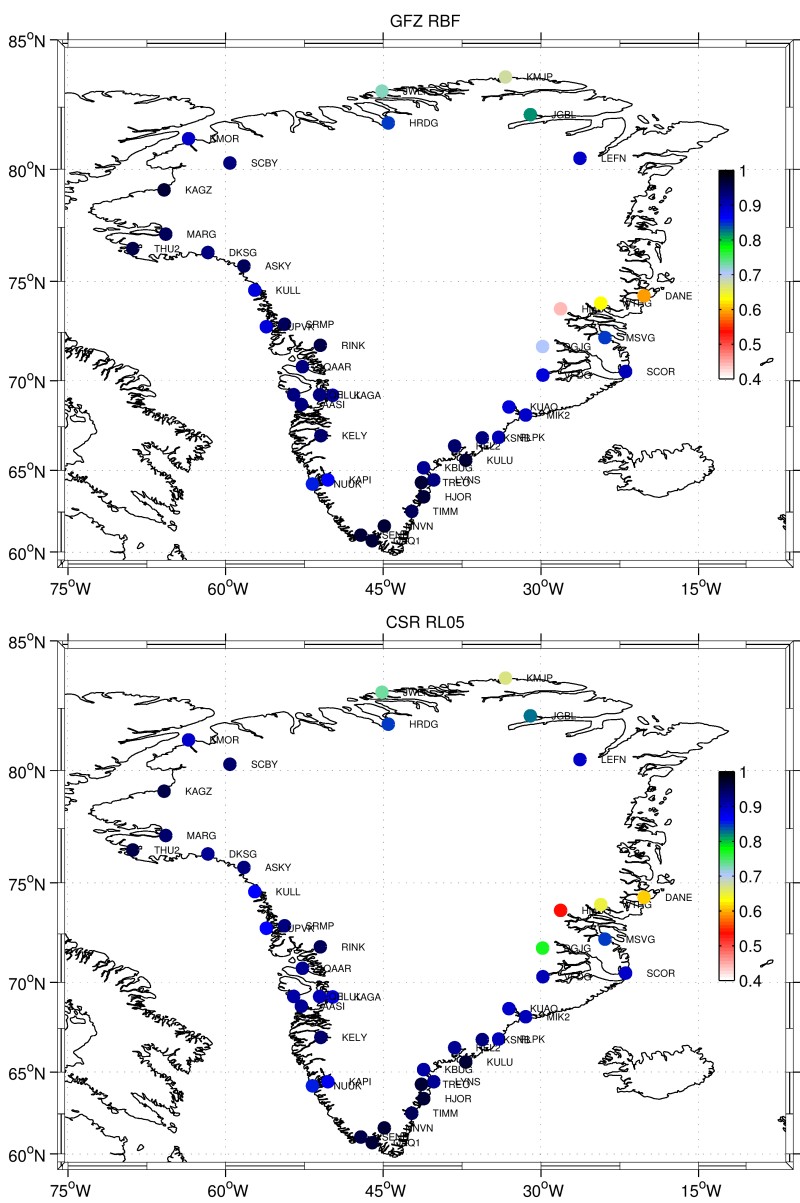

**Figure 1.** Correlations between the G-NET station uplift and the ice-induced crustal deformations due to dynamic loading of the crustal layer obtained using the temporal gravity field solutions. Only very minor differences for GFZ RBF and CSR RL5, mainly in the eastern part of Greenland can be exhibited.





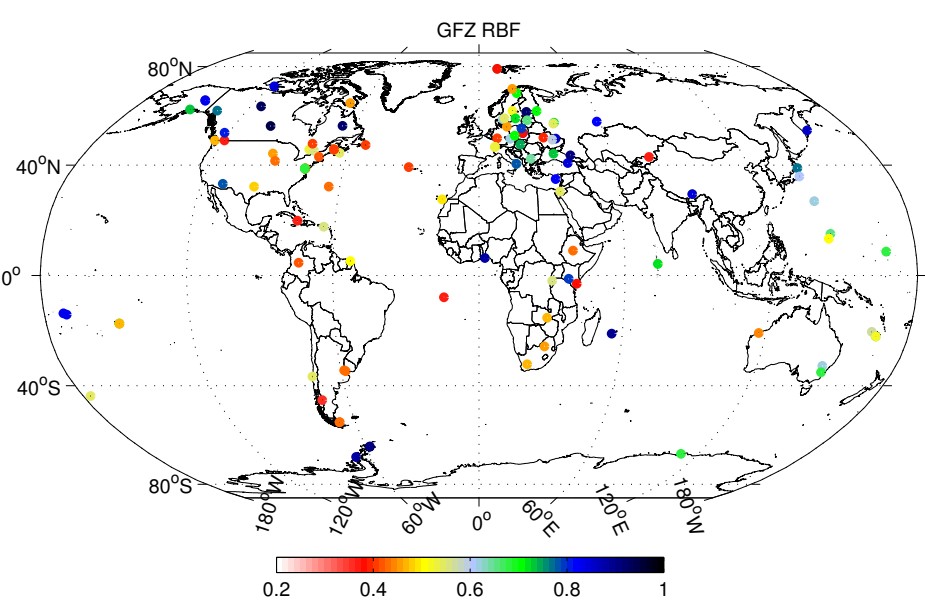

**Figure 2.** Correlations of the vertical station variations inferred from GFZ GRACE RBF solutions and the global GPS station network from CODE. Only stations with correlations $r > 0.2$ (in total 95 stations) were considered.

and serve as proxy for GFZ RL05a, ITSG2016 and CSR RL05, as well. Average correlations for the stations with correlations $r > 0.2$ (in total, 95 stations), are: CSR RL05: 56.8%, GFZ RBF: 56.6%, GFZ RL05a: 53.9% and ITSG2016: 53.5%.

The reason why the global station network generally correlates less than the G-NET sites can be explained by the uplift signal strength and the individual data quality (disruptions or damages) but also due to their location, e.g. on islands or coastal regions where signal separation is difficult. One should keep in mind that we are comparing (post-filtered) area mean values from GRACE with point values from GPS such that aliasing of neighboring signal occurs.

Nevertheless, for many stations the correlations are high (blue dots) and strongly support the ability of GRACE to remotely monitor mass induced uplift rates.

## 3   ICESAT AND GRACE MASS CHANGES

The extent of the Arctic sea ice has reached a new record low in September 2012. According to the European Environment Agency (2016), climate change causes sea ice melting in the



8    *Ch. Gruber et al.*

region at a rate much faster than estimated by earlier projections. The snow cover also shows a downward trend. The melting Arctic might impact not only the people living in the region, but thus also elsewhere in Europe and beyond.

Ice-mass changes of both the Greenland Ice Sheet (GIS) and the Antarctic Ice Sheet (AIS) have been inferred from monthly gravity fields of different GRACE solutions (GFZ RL05a, CSR RL05 and GFZ RBF). Except for GFZ RBF, all solutions have been filtered using an unisotropic decorrelating filter DDK4 (Kusche et al. 2009). Spherical harmonic degree 1 coefficients were added as described in Section 2 as well as the Earth's oblateness, $C_{2,0}$. Mass changes of the solid Earth due to glacial isostatic adjustment (GIA) have been corrected by means of the ICE-5G v1.3 model for the GIS and the IJ05_R2 model (Ivins et al. 2013) for the AIS. All results presented in the following are updates of the findings in Groh et al. (2014a, 2014b) to which the reader is referred for a detailed description of the processing.

Mass change time series for the GIS (01/2003–12/2013) are shown in Fig. 3. All time series are in good agreement and exhibit comparable linear and seasonal variations. Only minor differences are visible for specific periods. In general, the mass change time series for the AIS (Fig. 4) are also in good agreement. Although differences in the linear trend estimates are visible, they still agree with the corresponding accuracy measures, which are clearly dominated by remaining uncertainties in the GIA predictions.

ICESat laser altimetry observations can be used to derive linear ice-mass changes over Greenland and Antarctica, which can be compared to corresponding GRACE results. Here we utilise the ICESat-derived mass change estimates presented in Groh et al. (2014a, 2014b) to compare them to our GRACE ice-mass trend estimates for the period 10/2003–10/2009, the operational period of ICESat. Additional trend estimates for selected drainage basins are compared in Fig. 5. Despite the different observation techniques and resolution capabilities Fig. 5 reveals an overall good agreement between the tested solutions. Still, some differences between ICESat and the three GRACE solutions, exist. For example, the ICESat results for eastern Greenland exceed those from GRACE substantially. Moreover, while GRACE observes a mass gain for the East Antarctic Ice Sheet, the opposite conclusion can be drawn from the ICESat results. These differences can be related to the different error sources of both techniques. Moreover, limitations in the density assumption (here: density of pure ice) used to convert altimetric height changes into mass change can also contribute to the revealed differences.





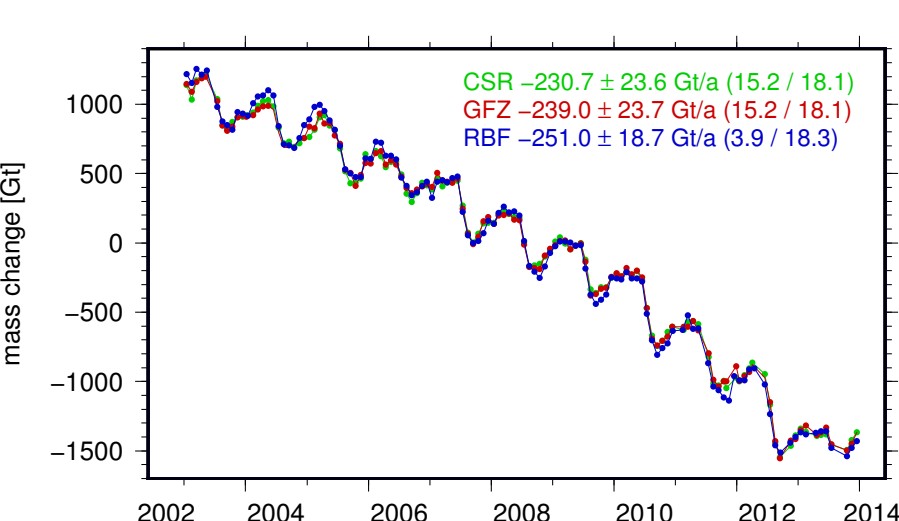

**Figure 3.** Greenland linear Ice Sheet mass change estimates per year from different GRACE solutions (CSR RL05, GFZ RL05a and GFZ RBF) from 01/2003 - 12/2013. Values in brackets indicate different components of the total error budget (GIA model uncertainties – last value and all remaining error contributions, including leakage errors and GRACE errors – first value).

## 4  GLOBAL MAJOR HYDROLOGICAL BASIN COMPARISON

Global catchment aggregated values (CAVs) for hydrological basins greater than $\approx 50,000\ km^2$ have been computed from WGHM (Döll et al., 2003) and compared to the equivalent water layer variations (EWH, according to Wahr et al. 1998) from results obtained from GRACE. The aggregation was performed by equally weighted sums over regular surface tiles.

The GRACE monthly fields were used after post processing with DDK4 according to Kusche et al. (2009), consistently for the spherical harmonic models (CSR RL5, GFZ RL05a and ITSG2016) and monthly mean values of daily Kalman filtered results for the GFZ RBF solution. The GRACE data have again been reduced for glacial isostatic adjustment (GIA) and seasonal variations were removed beforehand from all data sets in order to focus on non-





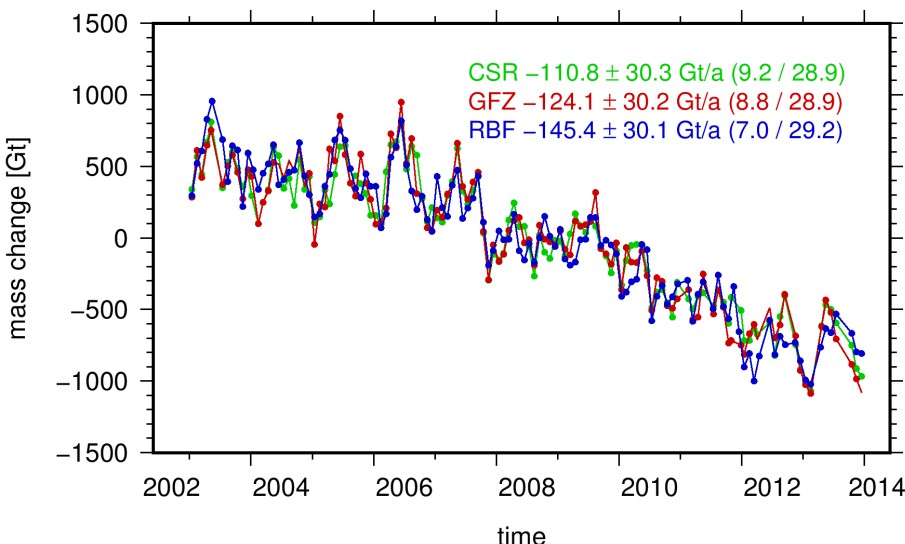

**Figure 4.** Antarctic linear Ice Sheet mass change estimates per year from different GRACE solutions (CSR RL05, GFZ RL05a and GFZ RBF) from 01/2003 - 12/2013. Values in brackets indicate different components of the total error budget (GIA model uncertainties – last value and all remaining error contributions, including leakage errors and GRACE errors – first value).

seasonal coherence. Moreover, in the case of the GFZ RBF solution, the seasonal cycle has already been introduced as a time variable background model.

The database containing in total 188 basins (of which 163 were used) was obtained from the interactive GeoNetwork (FAO, 2015). We used (i) Pearson's bi-variate correlation coefficient (XO), (ii) the standard deviation (SD) of the differences between two series and (iii) the scale corresponding to the GRACE basin series w.r.t it's hydrological counterpart, in order to reveal their agreement. The averaged agreements, are displayed in Tab.1. A positive correlation threshold of 10% was presumed for the individual GRACE solutions for each basin to exclude e.g. deserts or islands, where strong impact from signal leakage of surrounding water deteriorates our results.



**Table 1.** Comparison of average GRACE basin estimates against hydrological modeling (WGHM). Bold faced numbers highlight the best performance in the category. Values in brackets are obtained if the seasonal signal is included.

|               | ITSG2016      | GFZ RL05a    | GFZ RBF         | CSR RL5       |
| ------------- | ------------- | ------------ | --------------- | ------------- |
| (i) XO [%]    | **60.5 (70.6)** | 53.1 (65.6) | 53.1 (68.2)    | 57.4 (68.5)   |
| (ii) SD(d) [cm] | 4.45 (7.44) | 4.81 (7.68)  | **4.37 (7.36)** | 4.64 (7.56)   |
| (iii) Scale   | 0.93 (0.98)   | 0.90 (0.96)  | **0.96 (1.00)** | 0.92 (0.97)   |

All four solutions perform very close with only minor differences, mainly discovered in terms of correlations to the hydrology model (WGHM) over the timespan (2002-2013) . While the correlation gives an opportunity to find out how coherent our remotely sensed results represent a certain 'ground truth', the SD of the differences indicate the reliability of the results. The amplitudes indicate to which extent remote mass balances are captured on average.

Best correlation results have been found for the ITSG 2016 solution with 60.5% for the de-seasoned results and 70.6% for the full signal. Lowest standard deviations of the differences to hydrological basin averages were found with 4.4 cm for GFZ RBF after de-seasoning and 7.4 cm for the full signal. The best scale correspondence which projects GRACE basin estimates onto the reference hydrology were found for the GFZ RBF solutions. GRACE equivalent water layer estimates thus capture on average most of the hydrological signal strength.

Fig. 7 displays the comparative correlations for each basin w.r.t. the hydrological model (WGHM) that represents total water storage variations throughout the period $2002 - 2013$. This comparative comparison provides a performance indicator for different GRACE solutions by means of their individual agreement with WGHM on the level of CAVs and geographical location.

Still, remains difficult to identify systematic patterns such as basin size or basin location that would indicate e.g. data sampling or specific processing properties. The results overall strongly support the capability of GRACE to monitor global water storage variations remotely from space despite of the band limitation of the solutions and their signal omission errors.

To counteract this in Steckler et al. (2010), the basin scale masks for water loading in Bangladesh were processed by a truncated spherical harmonic representation in order to simulate the omission error from the model resolution. In our approach we have converted each





fine scale basin mask of $[0.5° \times 0.5°]$ into a coarse mask of $[2° \times 2°]$ which entirely includes the fine scale mask in the sense of a convex hull. The domain is thus enlarged to encounter to a certain extent for signal leakage-out effects. On the other hand leakage-in cannot be treated effectively other than by an increased model resolution under the provision that the measurement system is sensitive to it. Main limitations thus remain gravity signal attenuation at GRACE mission altitude and the separation width of the twin satellite system.

## 5    ALTIMETRY SATELLITE ORBITS

Recently, the impact of time variable geopotential models on altimetry satellite orbits has been investigated (Rudenko et al. 2014). Following these ideas, we test the GFZ RBF solutions for precise orbit determination of Envisat (2002-2012), Jason-1 (2002-2013) and Jason-2 (2008-2015) at the time intervals given in the parentheses.

We have chosen these satellites since their missions coincide with the GRACE time interval. The orbits are derived at 7-day arcs for Envisat and 12-day arcs for Jason-1 and Jason-2 by using the same background models for each satellite (Rudenko et al. 2017), but choosing three different Earth gravity field models/solutions: EIGEN-6S4 (Förste et al. 2016), GFZ RBF and GFZ RL05a. For the propagation of the orbits, based on the GFZ RBF time variable part, we first convert the grid tiles into spherical harmonic coefficients, and add the static part of the EIGEN-6S4 model. The static part of the satellite-only global gravity field model EIGEN-6S4 is complete up to degree and order 300. The time variable gravity part of the model is represented by a drift, annual and semi-annual variations per year of spherical harmonic coefficients up to degree and order 80 by July 1, 2014.

We have computed fits (observed minus calculated) of SLR and DORIS observations used for precise orbit determination of the satellites and two-day arc overlaps. Since the only difference in our tests consists in a replacement of Earth's gravity field models/solutions, smaller values of observation fits and arc overlaps indicate better performance of a respective Earth's gravity field model/solution.

The mean values of SLR and DORIS RMS fits and two-day radial arc overlaps for each satellite obtained using the EIGEN-6S4 model, GFZ RL05a and GFZ RBF solutions are shown in Tab. 2.





**Table 2.** The mean values of SLR and DORIS RMS fits and two-day radial arc overlaps for Envisat (2002-2012), Jason-1 (2002-2013) and Jason-2 (2008-2015), obtained using the EIGEN-6S4 model, GFZ RBF and GFZ RL05a solutions.

| Satellite | Altitude [km] | Model/ Solution | SLR fits [cm] | DORIS fits [mm/s] | Radial arc overlap [cm] |
|---|---|---|---|---|---|
| Envisat | 800 | EIGEN-6S4 | 1.27 | 0.4214 | 0.53 |
| | | GFZ RBF | 1.28 | 0.4215 | 0.57 |
| | | GFZ RL05a | 1.28 | 0.4216 | 0.60 |
| Jason-1 | 1336 | EIGEN-6S4 | 1.19 | 0.3532 | 0.79 |
| | | GFZ RBF | 1.20 | 0.3538 | 0.77 |
| | | GFZ RL05a | 1.19 | 0.3533 | 0.79 |
| Jason-2 | 1336 | EIGEN-6S4 | 1.23 | 0.3486 | 0.56 |
| | | GFZ RBF | 1.24 | 0.3486 | 0.56 |
| | | GFZ RL05a | 1.23 | 0.3489 | 0.56 |

The results obtained using the GFZ RBF solutions are in agreement with those obtained using the EIGEN-6S4 model and slightly outperform the results obtained using the GFZ RL05a solution. Since Envisat is more sensitive to the Earth's gravitational field due to its lower altitude than two Jason satellites, we look at the results obtained for this satellite in more detail. The DORIS measurements (Fig. 8) seem to be less suitable to detect the impact of the replacement of EIGEN-6S4 gravity field model by GFZ RBF solutions, since there are no notable differences in the fits of these observations derived different Earth's gravity field realizations.

SLR RMS fits (Fig. 9) show comparable or even better performance (smaller RMS fits) at some orbital arcs for Envisat until the middle of 2008 when using GFZ RBF solutions and better performance when using the EIGEN-6S4 model from the middle of 2008 onwards. This is probably caused by insufficient trend estimates in the background modeling and can be addressed in a next iteration. The inconsistency is also confirmed when looking at weekly obtained two-day arc overlaps in Fig. 10. The radial arc overlaps are of comparable accuracy when using GFZ RBF, GFZ RL05a solutions and the EIGEN-6S4 model for Jason-1 and Jason-2, while for Jason-1, the GFZ RBF solutions even outperform the model and other solutions, cf. Table 1.



## 6    DISCUSSION AND OUTLOOK

In this study a set of evaluation methods is used to compare the novel RBF GRACE solutions with other widely used standard GRACE solutions. Their absolute figures confirm once again the high potential and ability of GRACE or GRACE-like missions to significantly contribute to climate relevant indicators such as the quantification of ice-mass loss over Greenland. While a single correlation result gives only limited evidence of the overall quality of a solution, the sum over several evaluations may provide a fair picture of the relative performances in a close comparison with each other. The obtained spread of results is found relatively small and has clearly converged with each new release, however, still minor differences are found and may help to further improve the data processing methods within the GRACE community.

More in detail, the comparison to G-NET and CODE GPS uplift rates confirmed the temporal loading of mass redistribution that is revealed in the GRACE solutions. Both vertical data sets have helped in the past to validate and confirm the spatial resolution of the GRACE results. All four GRACE time variable gravity field solutions that we have tested (ITSG2016, GFZ RL05a, GFZ RBF and CSR RL05) show consistently high correlations (89-90%) with the vertical site displacements from the G-NET GPS Network. The correlations to the global GPS station network from CODE are lesser (52-55%). This can be explained by the lower uplift signal strength and the individual data quality but also due to their location, e.g. on islands. However, for many stations the correlations are high and confirm the ability of GRACE to remotely monitor mass induced uplift rates.

Our direct comparison with linear ice-mass changes from ICESat results with the GRACE loading data reveals a very good agreement, but also spatial differences, when comparing over smaller drainage basins.

The comparative agreement to the hydrological model WGHM shows that monthly means of the GFZ RBF solutions are of equal quality as the renowned products. All GRACE models under consideration perform very closely and support the fact that large scale hydrology can be accurately monitored remotely from space, especially the trend estimates of the Earth's polar ice-sheets melting and groundwater depletion over large deserted areas. The transformation of K-Band and trajectory data from dynamic to in-situ observations has been successfully used to compute the GFZ RBF solutions. An improved de-aliasing for monthly gravity field products is feasible when estimating additional sub-monthly results for time-variable gravity signals and residual atmosphere and oceanic loading. The (Kalman-) regularization reduces artifacts during inversion such that no post-filtering is indicated for these products.

Precise orbit determination of low orbit Earth's satellites, such as e.g. Envisat, has been



shown to be a powerful tool to validate daily and monthly Earth's time-variable gravity field solutions. In general, the orbit tests for altimetry satellites Envisat, Jason-1 and Jason-2 over the total 2002-2015 time interval show rather comparable quality of the orbits derived using EIGEN-6S4 model, GFZ RBF and GFZ RL05a solutions. DORIS measurements seem to be less sensitive to the replacement of up-to-date time variable Earth gravity field models and solutions. On the contrary, SLR residuals and arc overlaps of altimetry satellite orbits are sensitive to the quality of the underlying background models. From 2002 until the middle of 2008, SLR RMS fits of Envisat obtained using GFZ RBF solutions perform comparably and even better at some weeks than those derived using the EIGEN-6S4 model, whereas this model outperforms the GFZ RBF solutions from 2008 onwards.

Radial arc overlaps are of comparable accuracy, when using GFZ RBF, GFZ RL05a solutions and the EIGEN-6S4 model for Jason-1 and Jason-2, while for Jason-1, the GFZ RBF solutions even outperform the model and other solutions. For Envisat, which is more sensitive to the gravity field modeling, the smallest radial arc overlaps are obtained using the EIGEN-6S4 model, followed by GFZ RBF solutions and finally by GFZ RL05a solutions. In this context, future reprocessing of GRACE time series can be verified against altimetry results to confirm further improvements. In view of an upcoming GRACE follow-On mission with improved instrument data, we may expect time-variable gravity fields to be included in future orbit computations of altimetry satellites.

## ACKNOWLEDGMENTS

We would like to thank the German Space Operations Center (GSOC) of the German Aerospace Center (DLR) for providing continuous and nearly 100% of the raw telemetry data of the twin GRACE satellites. The WGHM hydrological data sets have been greatly appreciated. We would like also to thank the CODE processing team for providing the CODE data as well as the Greenland GPS station network for providing the G-NET vertical displacements. Ryan L. Sink is thanked for English lecturing.

This research was partly supported by the European Space Agency (ESA) within the Climate Change Initiative Sea Level Phase 2 project and by the German Research Foundation (DFG) within the project "Consistent dynamic satellite reference frames and terrestrial geodetic datum parameters". SLR and DORIS data available from the International Laser Ranging Service (ILRS) and International DORIS Service (IDS) were used in this research. One of the authors (Ch. Gruber) was funded by the European Union's Horizon 2020 project European Gravity Service for Improved Emergency Management (EGSIEM) under the grant





agreement No 637010. This article reflects only the authors' views. The Research Executive
Agencies are not responsible for any use that may be made of the information it contains.

Latest daily $[2° \times 2°]$ grids in equivalent water heights, and $[1° \times 1°]$ grids with GIA predic-
tions removed and center of mass to center of figure corrected, as well as spherical harmonic
coefficients, can be downloaded from

`ftp://gfzop.gfz-potsdam.de/EGSIEM/`.

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





**Table A1.** List of global hydrological catchment basins sorted by area. The correlations refer to the signal from GRACE derived water storage variations and the hydrological model WGHM after deseasoning. The given numbers are a performance indicator for the individual solutions in a relative context.

| Basin Name | Size [km$^2$] | ITSG**2016** [%] | GFZ **RL05a** [%] | GFZ **RBF** [%] | CSR **05** [%] |
|---|---|---|---|---|---|
| Narva | 48838 | 74 | 66 | 51 | 69 |
| StJohn | 55210 | 74 | 68 | 85 | 74 |
| SouthPacificIslands | 58689 | 24 | 36 | 26 | 26 |
| EmsWeser | 65326 | 49 | 30 | 19 | 47 |
| ItalyWestCoast | 68891 | 70 | 52 | 58 | 63 |
| Guadiana | 70409 | 78 | 72 | 82 | 76 |
| Tagus | 72920 | 79 | 73 | 84 | 78 |
| Dniester | 73438 | 67 | 65 | 75 | 67 |
| Gironde | 80159 | 85 | 75 | 80 | 77 |
| Farahrud | 82474 | 26 | 26 | 14 | 21 |
| IndiaWestCoast | 84089 | 73 | 68 | 68 | 70 |
| BayofBengalNorthEastCoast | 85714 | 92 | 90 | 90 | 91 |
| Ireland | 85904 | 33 | 16 | 14 | 37 |
| Daugava | 86070 | 85 | 81 | 77 | 84 |
| Neman | 92930 | 76 | 69 | 62 | 68 |
| ItalyEastCoast | 92978 | 82 | 69 | 73 | 77 |
| SpainPortugalAtlanticCoast | 93024 | 79 | 67 | 83 | 77 |
| Churchill | 93099 | 62 | 63 | 82 | 63 |
| Douro | 97412 | 81 | 72 | 82 | 80 |
| Narmada | 98279 | 80 | 80 | 78 | 81 |
| Rhone | 98367 | 81 | 76 | 73 | 77 |
| SouthAfricaWestCoast | 102100 | 46 | 42 | 61 | 35 |
| SpainSouthandEastCoast | 102185 | 64 | 52 | 53 | 60 |
| BalticSeaCoast | 106081 | 59 | 51 | 43 | 50 |
| FranceWestCoast | 108390 | 79 | 70 | 87 | 72 |
| Loire | 117049 | 84 | 80 | 81 | 81 |
| Oder | 121292 | 72 | 67 | 74 | 73 |
| CentralPatagoniaHighlands | 121293 | 55 | 45 | 52 | 58 |
| MarChiquita | 129715 | 68 | 69 | 64 | 65 |
| RioLerma | 130820 | 78 | 74 | 79 | 73 |
| GrijalvaUsumacinta | 132049 | 92 | 88 | 91 | 91 |
| Elbe | 140922 | 77 | 66 | 70 | 78 |
| Mahandi | 144672 | 88 | 87 | 84 | 88 |
| RussiaSouthEastCoast | 150259 | 67 | 61 | 51 | 59 |
| RioBalsas | 156042 | 77 | 71 | 74 | 76 |
| ChaoPhraya | 157686 | 91 | 91 | 90 | 90 |
| Negro | 162658 | 73 | 71 | 79 | 73 |
| HongRedRiver | 165007 | 86 | 86 | 82 | 84 |
| PampasRegion | 175610 | 35 | 35 | 21 | 33 |
| SalinasGrandes | 177187 | 49 | 52 | 46 | 48 |
| HamuniMashkel | 179360 | 71 | 62 | 65 | 60 |
| NorthandSouthKorea | 181759 | 49 | 41 | 64 | 42 |
| VietNamCoast | 186187 | 94 | 92 | 93 | 93 |
| Rhine | 187991 | 78 | 64 | 67 | 73 |
| EasternJordanSyria | 189266 | 40 | 33 | 17 | 39 |
| Sulawesi | 190307 | 77 | 68 | 73 | 77 |
| ArabianSeaCoast | 190641 | 73 | 54 | 72 | 57 |
| Wisla | 193658 | 77 | 72 | 75 | 72 |
| YucatanPeninsula | 197472 | 91 | 86 | 89 | 90 |
| NorthBorneoCoast | 202997 | 59 | 53 | 52 | 53 |
| PersianGulfCoast | 207160 | 52 | 44 | 45 | 49 |
| Ural | 215178 | 70 | 68 | 75 | 68 |
| JavaTimor | 223696 | 67 | 60 | 44 | 64 |
| NorthArgentinaSouthAtlanticCoast | 224076 | 65 | 64 | 56 | 67 |
| Neva | 229621 | 81 | 78 | 74 | 80 |
| Fraser | 232176 | 87 | 86 | 84 | 87 |
| Caribbean | 232942 | 69 | 61 | 55 | 71 |
| MexicoInterior | 239690 | 68 | 59 | 67 | 61 |
| AfricaIndianOceanCoast | 244531 | 73 | 65 | 54 | 70 |
| Helmand | 250573 | 57 | 52 | 54 | 53 |
| UnitedStatesNorthAtlanticCoast | 255343 | 77 | 64 | 81 | 71 |
| Magdalena | 259632 | 75 | 62 | 63 | 69 |
| NamibiaCoast | 260457 | 47 | 34 | 13 | 35 |
| BlackSeaNorthCoast | 262302 | 74 | 68 | 67 | 74 |
| Salween | 265822 | 88 | 88 | 85 | 88 |
| NorthBrazilSouthAtlanticCoast | 271751 | 89 | 86 | 87 | 88 |
| NewZealand | 272526 | 32 | 26 | 61 | 29 |
| Krishna | 274198 | 80 | 82 | 80 | 80 |
| NorthernDvina | 274880 | 95 | 93 | 92 | 95 |
| EastBrazilSouthAtlanticCoast | 285877 | 83 | 82 | 90 | 81 |
| Finland | 290606 | 69 | 60 | 55 | 65 |
| ColombiaEcuadorPacificCoast | 290939 | 34 | 23 | 35 | 24 |
| PeruPacificCoast | 290939 | 44 | 37 | 42 | 40 |
| PapuaNewGuineaCoast | 291136 | 67 | 62 | 62 | 68 |
| Philippines | 304285 | 73 | 49 | 76 | 66 |





| Basin Name | Size [km²] | ITSG 2016 [%] | GFZ RL05a [%] | GFZ RBF [%] | CSR RL05 [%] |
|---|---|---|---|---|---|
| PeninsulaMalaysia | 311477 | 16 | 20 | 15 | 16 |
| Godavari | 313892 | 87 | 86 | 84 | 87 |
| CaribbeanCoast | 317043 | 85 | 80 | 81 | 84 |
| BlackSeaSouthCoast | 318639 | 64 | 59 | 58 | 65 |
| Parnaiba | 331643 | 95 | 93 | 96 | 93 |
| AdriaticSeaGreeceBlackSeaCoast | 342127 | 82 | 83 | 87 | 82 |
| MediterraneanSeaEastCoast | 342785 | 76 | 73 | 83 | 77 |
| LaPunaRegion | 348890 | 79 | 62 | 47 | 73 |
| BoHaiKoreanBayNorthCoast | 353244 | 62 | 57 | 48 | 63 |
| GreatBasin | 370144 | 84 | 82 | 78 | 80 |
| CaspianSeaSouthWestCoast | 371831 | 31 | 38 | 40 | 31 |
| SouthAmericaColorado | 373863 | 48 | 45 | 47 | 50 |
| Japan | 378301 | 44 | 47 | 18 | 43 |
| SouthernCentralAmerica | 387927 | 91 | 89 | 90 | 89 |
| Irrawaddy | 402028 | 93 | 92 | 92 | 92 |
| SouthAfricaSouthCoast | 403126 | 72 | 47 | 48 | 69 |
| Volta | 411058 | 75 | 71 | 69 | 70 |
| Limpopo | 411553 | 59 | 45 | 40 | 56 |
| XunJiang | 412953 | 87 | 86 | 90 | 86 |
| California | 420022 | 78 | 73 | 78 | 76 |
| Don | 445212 | 66 | 66 | 69 | 62 |
| LakeBalkash | 445594 | 73 | 67 | 71 | 68 |
| IrianJayaCoast | 449015 | 73 | 66 | 58 | 65 |
| GulfCoast | 465689 | 86 | 84 | 74 | 85 |
| Senegal | 477345 | 93 | 85 | 91 | 91 |
| Sumatra | 477814 | 41 | 28 | 18 | 33 |
| MexicoNorthwestCoast | 478301 | 69 | 56 | 70 | 69 |
| SouthArgentinaSouthAtlanticCoast | 484180 | 51 | 48 | 63 | 53 |
| Sweden | 489477 | 75 | 71 | 69 | 79 |
| AustraliaSouthCoast | 490397 | 34 | 34 | 45 | 30 |
| AngolaCoast | 499542 | 74 | 66 | 62 | 70 |
| Dnieper | 513535 | 74 | 72 | 72 | 75 |
| Sabarmati | 523530 | 61 | 59 | 49 | 58 |
| Kalimantan | 542536 | 75 | 68 | 71 | 71 |
| RioGrandeBravo | 552385 | 57 | 50 | 46 | 51 |
| MediterraneanSouthCoast | 558292 | 26 | 12 | 18 | 27 |
| CaspianSeaCoast | 561343 | 65 | 65 | 63 | 64 |
| NortheastSouthAmericaSouthAtlanticCoast | 561413 | 80 | 79 | 70 | 80 |
| ScandinaviaNorthCoast | 578748 | 84 | 74 | 64 | 79 |
| Madasgacar | 596220 | 88 | 78 | 86 | 84 |
| SaoFrancisco | 635159 | 90 | 86 | 87 | 87 |
| RiftValley | 638878 | 56 | 42 | 38 | 53 |
| NorthAmericaColorado | 650155 | 72 | 65 | 71 | 68 |
| ChinaCoast | 650882 | 74 | 58 | 60 | 74 |
| RussiaBarentsSeaCoast | 678113 | 96 | 91 | 88 | 95 |
| AtlanticOceanSeaboard | 689995 | 78 | 72 | 90 | 79 |
| KaraSeaCoast | 696301 | 89 | 87 | 89 | 88 |
| GulfofGuinea | 699755 | 36 | 34 | 28 | 36 |
| GulfofMexicoNorthAtlanticCoast | 701385 | 83 | 81 | 75 | 84 |
| AustraliaEastCoast | 734552 | 76 | 66 | 68 | 69 |
| AustraliaWestCoast | 738000 | 31 | 37 | 61 | 29 |
| ColumbiaandNorthwesternUnitedStates | 757681 | 90 | 89 | 86 | 89 |
| CentralIran | 787176 | 44 | 34 | 22 | 42 |
| ShebelliJuba | 796599 | 53 | 37 | 49 | 55 |
| AmuDarya | 799261 | 83 | 78 | 81 | 83 |
| Danube | 799650 | 81 | 82 | 88 | 81 |
| Mekong | 803303 | 90 | 91 | 88 | 89 |
| AfricaNorthWestCoast | 809724 | 29 | 22 | 21 | 28 |
| UruguayBrazilSouthAtlanticCoast | 830359 | 76 | 73 | 68 | 71 |
| HuangHe | 832494 | 39 | 44 | 35 | 34 |
| AfricaSouthInterior | 863869 | 81 | 79 | 71 | 81 |
| Indus | 867157 | 53 | 58 | 55 | 54 |
| Tocantins | 915661 | 94 | 94 | 95 | 93 |
| TigrisEuphrates | 916137 | 71 | 73 | 72 | 71 |
| MurrayDarling | 928776 | 76 | 70 | 85 | 72 |
| Orinoco | 974772 | 93 | 93 | 92 | 93 |
| Orange | 984867 | 76 | 59 | 40 | 69 |
| AfricaWestCoast | 1010044 | 85 | 85 | 84 | 85 |
| AfricaEastCentralCoast | 1041192 | 78 | 78 | 73 | 78 |



| Basin Name | Size [km²] | ITSG 2016 [%] | GFZ RL05a [%] | GFZ RBF [%] | CSR RL05 [%] |
|---|---|---|---|---|---|
| SyrDarya | 1117625 | 75 | 69 | 71 | 77 |
| SaskatchewanNelson | 1135754 | 62 | 59 | 56 | 63 |
| SiberiaNorthCoast | 1200168 | 89 | 83 | 87 | 85 |
| StLawrence | 1309589 | 75 | 77 | 87 | 76 |
| Zambezi | 1373296 | 90 | 90 | 88 | 90 |
| Volga | 1474073 | 84 | 85 | 87 | 85 |
| HudsonBayCoast | 1648738 | 29 | 34 | 62 | 28 |
| GangesBramaputra | 1671358 | 74 | 74 | 67 | 74 |
| AustraliaNorthCoast | 1692704 | 93 | 93 | 91 | 93 |
| Mackenzie | 1766094 | 85 | 82 | 82 | 82 |
| Yangtze | 1789482 | 85 | 80 | 81 | 83 |
| Amur | 2086009 | 70 | 68 | 54 | 71 |
| Niger | 2136941 | 92 | 90 | 89 | 91 |
| ArcticOceanIslands | 2166086 | 13 | 13 | 17 | 14 |
| GobiInterior | 2170053 | 53 | 28 | 28 | 44 |
| PacificandArcticCoast | 2266165 | 65 | 60 | 64 | 66 |
| Lena | 2416437 | 76 | 74 | 76 | 74 |
| LakeChad | 2461890 | 86 | 82 | 91 | 86 |
| Yenisey | 2574501 | 90 | 81 | 86 | 86 |
| LaPlata | 3016800 | 88 | 85 | 81 | 86 |
| Ob | 3025660 | 86 | 85 | 83 | 86 |
| NorthwestTerritories | 3044095 | 80 | 78 | 85 | 80 |
| AustraliaInterior | 3048596 | 69 | 68 | 43 | 68 |
| SiberiaWestCoast | 3052334 | 87 | 83 | 86 | 85 |
| Nile | 3074955 | 82 | 76 | 70 | 81 |
| MississippiMissouri | 3273240 | 83 | 81 | 78 | 84 |
| Congo | 3696670 | 66 | 57 | 65 | 64 |
| Amazon | 5970775 | 90 | 90 | 90 | 90 |





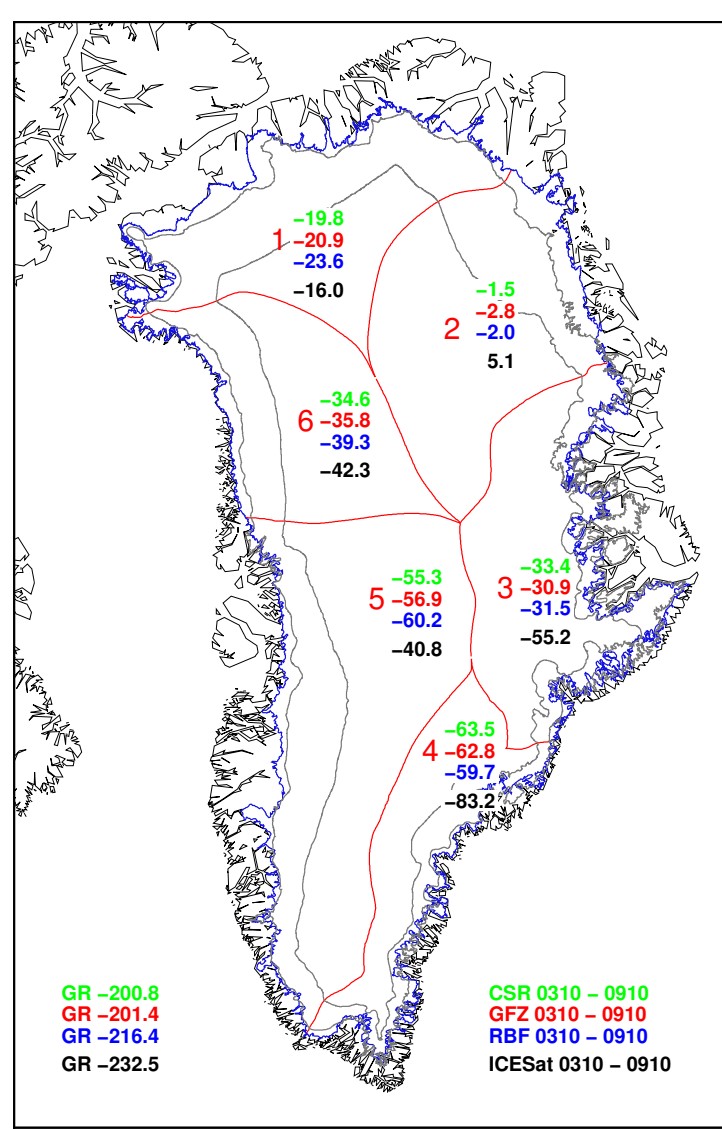

**Figure 5.** Mean annual ice-mass change (Gt/a) for the Greenland Ice Sheet as well as selected drainage basins (separated by red lines) and aggregations derived from different GRACE solutions and ICESat laser altimetry data over the period 10/2003–10/2009.

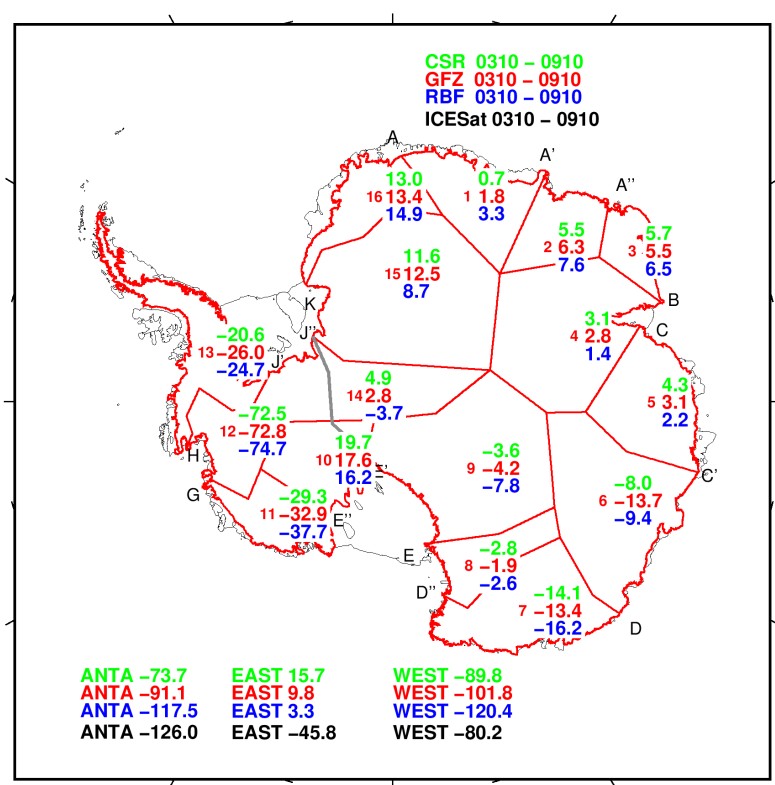

**Figure 6.** Mean annual ice-mass change (Gt/a) for both the Antarctic Ice Sheet as well as selected drainage basins (separated by red lines) and aggregations derived from different GRACE solutions and ICESat laser altimetry data over the period 10/2003–10/2009. The grey line depicts the boundary between the eastern and the western part of the AIS.





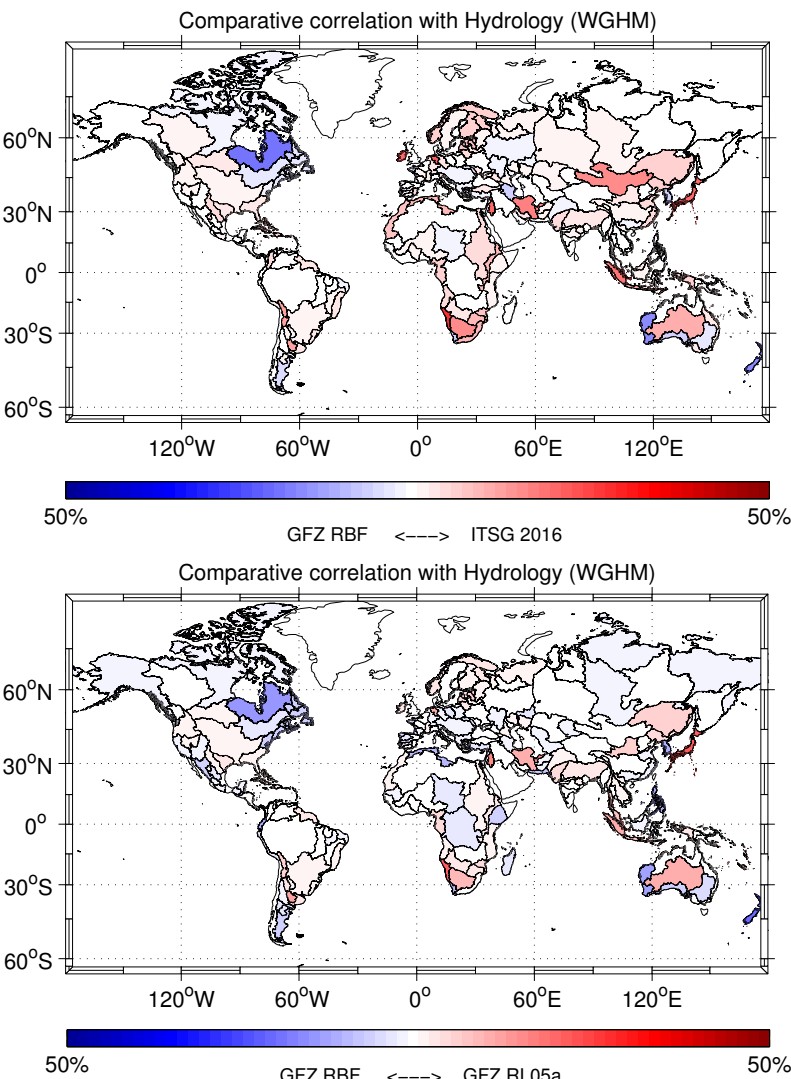

**Figure 7.** Comparative correlations of catchment aggregated values from GRACE results against hydrology; the plot depicts the relative difference (%) in each basin between the correlations of two time series with respect to the hydrological model (WGHM). Blue means higher coherence for the GFZ RBF solution and red marks higher coherence for the concurring model (GFZ RL05a, ITSG 2016). Hudson Bay Coast and Japan stick out slightly, which hints to post glacial rebound and the Tohoku megathrust Earthquake. See also text for further discussion. For a full list of all considered basins and their individual hydrological correlations, the reader is referred to Tab. A1 in the appendix.



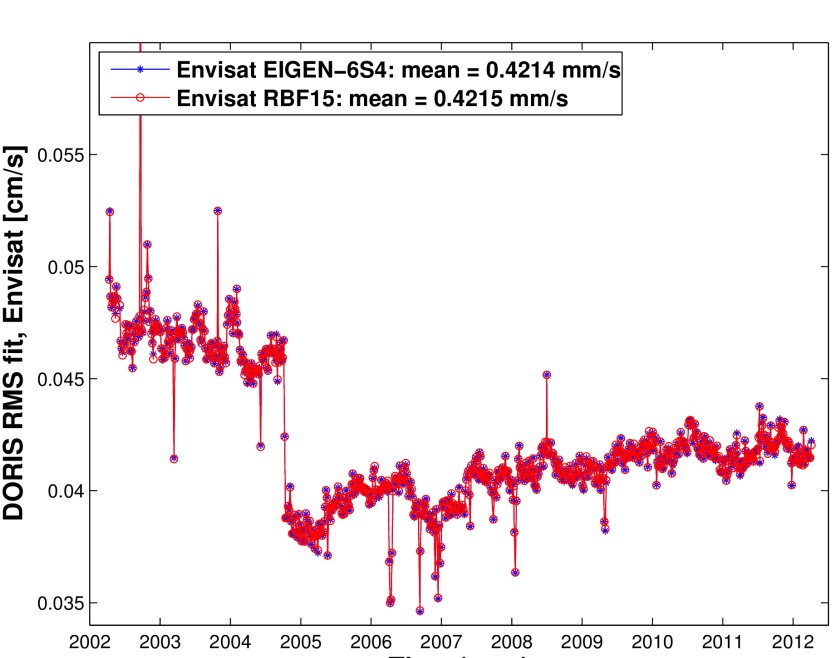

**Figure 8.** Weekly DORIS RMS fits of Envisat computed with different time-variable Earth gravity modeling: EIGEN-6S4 model and GFZ RBF solution.





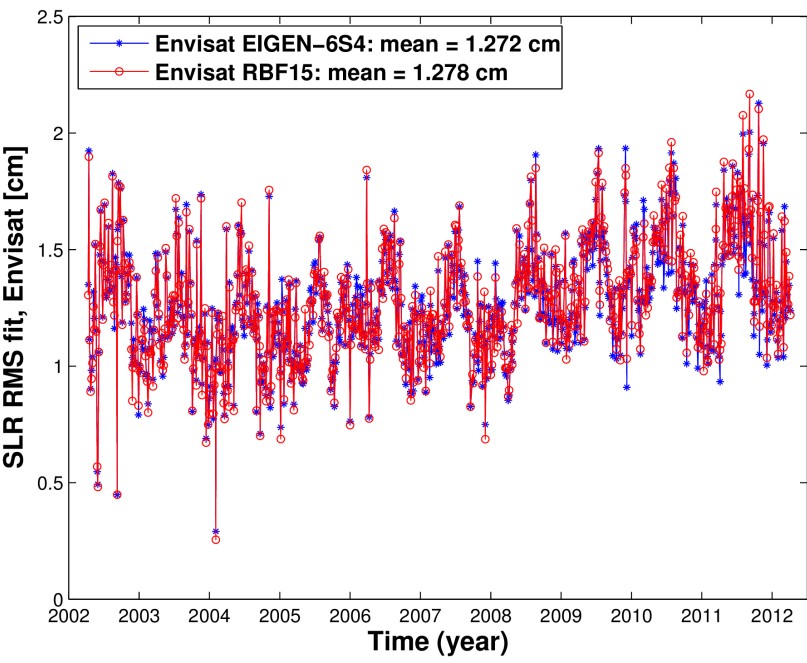

**Figure 9.** Weekly SLR RMS fits of Envisat computed with different time-variable Earth gravity modeling: EIGEN-6S4 model and GFZ RBF solution.



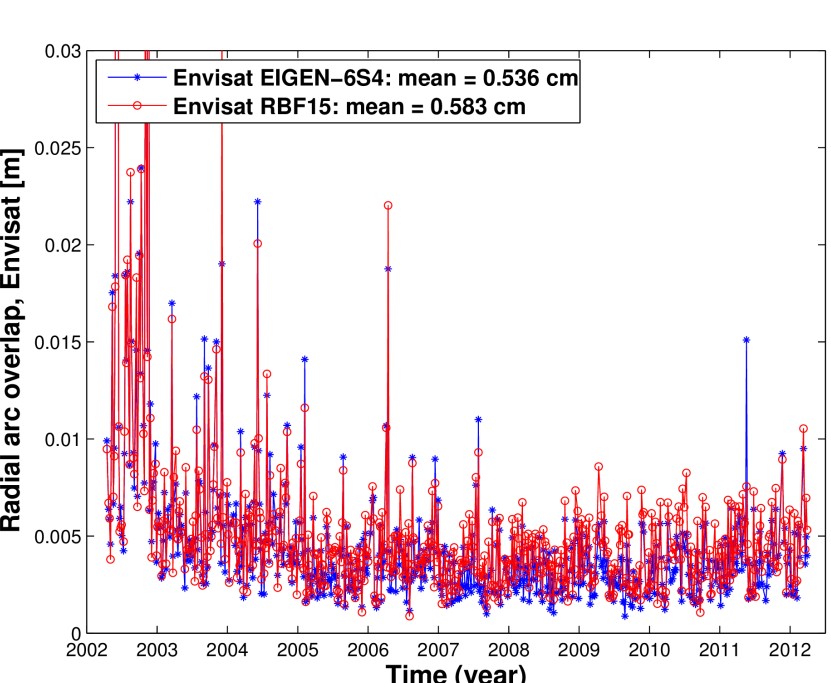

**Figure 10.** Weekly two-day radial arc overslaps for Envisat computed with different time-variable Earth gravity modeling: EIGEN-6S4 model and GFZ RBF solution.