# Peer review of "Earth's surface mass transport derived from GRACE, evaluated by GPS, ICESat, hydrological modeling and altimetry satellite orbits"

_Earth Surface Dynamics, 2017_

## Referee Comment (RC1) · Anonymous Referee #1 · 26 Jan 2018

Review on the manuscript "Earth's surface mass transport derived from GRACE, evaluated by GPS, ICESat, hydrological modeling and altimetry satellite orbits" (by Christian Gruber, Sergei Rudenko, Andreas Groh, Dimitrios Ampatzidis, Elisa Fagiolini) submitted to the journal "Earth Surface Dynamics"

The paper represents a possible approach of the combination of different observations for determination of the "surface mass transport". In my opinion the work is interesting in itself, but requires a revision in several place before the publication. Major remarks are reported in the following.

1. The paper basically refers to numerical results, which are quite valuable for such

regions as Greenland and Antarctic areas. Despite numerous illustrations, it is written rather in a compact way but may not be understood by the reader. For that reason this paper may be considered as a good paper, if authors will introduce a special section with the theory of determination of mass transport including all adopted assumptions (Probably authors supposed that surface masses are concentrated in a thin layer at the surface of the spherical Earth). If so, the following basic reference is missing:

Wahr, J., M. Molenaar, and F. Bryan (1998), Time variability of the Earth's gravity field: Hydrological and oceanic effects and their possible detection using GRACE, J. Geophys. Res., 103, 30,205– 30,229.

2. According to the title of manuscript, the authors should note that the determination of the mass redistribution at the Earth's surface from the given external potential is traditionally treated as special boundary case of improperly posed inverse problem of the gravitational potential. From this viewpoint accuracy estimates of the mass redistribution at the Earth's surface (Greenland, Antarctic) given in this work need to be explained in details. The authors can find the rigorous accuracy estimation for the case of such properly posed inverse problem in the following paper:

Marchenko A. N. (2009) On the Global Density Distribution Based on the Earth's Mechanical Parameters and Piecewise Reference Model. In: "Mission and Passion: Science" A volume dedicated to Milan Burša on the occasion of his 80th birthday , (Ed. P.Holota), Czech National Committee of Geodesy and Geophysics, Prague, 2009, pp. 169-179.

3. The reviewer has a certain doubts concerning the next remark (lines 12-14) of the authors: "In addition, non-conventional methods based on radial basis functions (RBF) and mascons will give the ability to compute models in regional and global representation as well". Such sentence could lead to a violation of understanding of the so-called RBF or radial multipole potentials (as harmonic functions) introduced by Maxwell (1881) and Lunar mascons (as singular objects) approximated by the potential of point

masses, disks, etc. On the other hand this sentence has direct relation rather to the modeling of the external potential then to the Earth's surface mass transport. Because of the Lauricella's theorem the direct modeling of second one is impossible without special study: any density distribution can be represented as simple sum of harmonic density and density of zero potential. See the basic reference:

Moritz, H. (1990) The Figure of the Earth. Theoretical Geodesy and Earth's Interior, Wichmann, Karlsruhe, 1990.

with the application of the Lauricella's theorem to the Earth's density

Marchenko A.N., Abrikosov O.V. A note on the standard parameterization of the Earth's dynamical figure and Darwin's density distribution. Bollettino di Geodesia et Scienze Affini, Anno LIX, (4), pp. 335-348

Summarizing, the reviewer recommends the publication of this manuscript in the Earth Surface Dynamics after improvements, including the Summary, according to given before suggestions.

Anonymous reviewer 26.01.2018

Please also note the supplement to this comment:
https://www.earth-surf-dynam-discuss.net/esurf-2017-70/esurf-2017-70-RC1-supplement.pdf

---

## Referee Comment (RC2) · Anonymous Referee #2 · 22 Mar 2018

The authors discuss a novel method based on radial basis functions for recovery of the global Earth's gravity field from GRACE inter-satellite range-rate data. To test its performance, they authors use four independent datasets and using various metrics they compare results of the new method with respect to three global geopotential models derived from GRACE data. Obtained numerical results demonstrate that the new method can be used for global gravity field modelling as an alternative to classical spherical (spheroidal) harmonic models.

In my opinion, the article contains interesting new results. Thus, I support the publication of the article in ESD, maybe after the authors consider my remarks:

[Figure]

1- Both the abstract and discussion should explicitly state that the RBF modelling technique can be used for processing GRACE data yielding global gravity field models which fit independent reference values at the same level as commonly accepted global geopotential models based on spherical harmonics. Advantages (and potentially also weaknesses) of the new technique (implementation complexity, computational cost, temporal and spatial resolvability etc.) could be mentioned in the text.

2- The RBF-based modelling technique is described in Section 1 using the standard text with some references. I would like to see at least fundamental equations of the mathematical model to make the article self-contained for its readers.

3- The alternative global geopotential models are referenced in Section 1 but for the same reason given above, few words about each model could be added (maximum resolution, processing technique, estimation of low-degree coefficients etc.).

---

## Author Comment (AC1) · 24 Apr 2018

We are thankful to the Reviewers for carefully reading our manuscript and their useful comments and suggestions that allowed us to improve the manuscript. In the following the comments of the Reviewers are given in black and our responses in blue.
Christian Gruber (on behalf of the co-authors)

Reviewer 1
The paper represents a possible approach of the combination of different observations for determination of the "surface mass transport". In my opinion the work is interesting in itself, but requires a revision in several place before the publication. Major remarks are reported in the following.

1. The paper basically refers to numerical results, which are quite valuable for such regions as Greenland and Antarctic areas. Despite numerous illustrations, it is written rather in a compact way but may not be understood by the reader. For that reason this paper may be considered as a good paper, if authors will introduce a special section with the theory of determination of mass transport including all adopted assumptions (Probably authors supposed that surface masses are concentrated in a thin layer at the surface of the spherical Earth). If so, the following basic reference is missing:

Wahr, J., M. Molenaar, and F. Bryan (1998), Time variability of the Earth's gravity field: Hydrological and oceanic effects and their possible detection using GRACE, J. Geophys. Res., 103, 30,205– 30,229.
We have added this citation.

line 62: used to solve for the gravity variations on surface grid tiles.
changed to:
used to solve for the surface mass equivalents that are concentrated in a thin layer at the surface of the spherical Earth (cf. Wahr et al. 1998).

2. According to the title of manuscript, the authors should note that the determination of the mass redistribution at the Earth's surface from the given external potential is traditionally treated as special boundary case of improperly posed inverse problem of the gravitational potential. From this viewpoint accuracy estimates of the mass redistribution at the Earth's surface (Greenland, Antarctic) given in this work need to be explained in details.
Added, line 71: It should be noted that this improperly posed inverse problem is constrained in two aspects.
Added before "Despite the regularized processing methodology,"
The formal accuracy estimates are found in the updated Kalman covariances that are epoch-wise co-estimated with the states. This results in the equivalent accuracy as

obtained from a regularized solution and is based on error propagation during the time update and a least-squares prediction error. For further details, the reader is referred to (Gruber et al., GJI in review).

The authors can find the rigorous accuracy estimation for the case
of such properly posed inverse problem in the following paper:
?? properly posed
Marchenko A. N. (2009) On the Global Density Distribution Based on the Earth's Mechanical Parameters and Piecewise Reference Model. In: "Mission and Passion: Science" A volume dedicated to Milan Burša on the occasion of his 80th birthday , (Ed. P.Holota), Czech National Committee of Geodesy and Geophysics, Prague, 2009, pp. 169-179.
We have added this citation.

3. The reviewer has a certain doubts concerning the next remark (lines 12-14) of the authors: "In addition, non-conventional methods based on radial basis functions (RBF) and mascons will give the ability to compute models in regional and global representation as well".
This is, in fact, what we show in our paper. We have deleted the word "will" in this sentence.

Such sentence could lead to a violation of understanding of the so-
called RBF or radial multipole potentials (as harmonic functions) introduced by Maxwell (1881) and Lunar mascons (as singular objects) approximated by the potential of point masses, disks, etc.

Yes, the Reviewer is right, we are only modeling the external potential of the Earth, condensed to the thin layer. Poisson's Equation is employed in exterior space. No misinterpretation can be expected here.

On the other hand this sentence has direct relation rather to the
modeling of the external potential then to the Earth's surface mass transport. Because of the Lauricella's theorem the direct modeling of second one is impossible without special study: any density distribution can be represented as simple sum of harmonic density and density of zero potential. See the basic reference:
Moritz, H. (1990) The Figure of the Earth. Theoretical Geodesy and Earth's Interior, Wichmann, Karlsruhe, 1990.
with the application of the Lauricella's theorem to the Earth's density
Marchenko A.N., Abrikosov O.V. A note on the standard parameterization of the Earth's dynamical figure and Darwin's density distribution. Bollettino di Geodesia et Scienze Affini, Anno LIX, (4), pp. 335-348

We believe, that this reference is of no relevancy as we have considered a thin layer of constant density, only. The main contributor to Earth density variations is removed as a static background model.

Summarizing, the reviewer recommends the publication of this manuscript in the Earth Surface Dynamics after improvements, including the Summary, according to given before suggestions.

Reviewer 2

The authors discuss a novel method based on (regularized) radial basis functions for recovery of the global Earth's gravity field from GRACE inter-satellite range-rate data. To test its performance, they authors use four independent datasets and using various metrics they compare results of the new method with respect to three global geopotential models derived from GRACE data. Obtained numerical results demonstrate that the new method can be used for global gravity field modelling as an alternative to classical spherical (spheroidal) harmonic models.

In my opinion, the article contains interesting new results. Thus, I support the publication of the article in ESD, maybe after the authors consider my remarks:

1- Both the abstract and discussion should explicitly state that the RBF modelling technique can be used for processing GRACE data yielding global gravity field models which fit independent reference values at the same level as commonly accepted global geopotential models based on spherical harmonics. Advantages (and potentially also weaknesses) of the new technique (implementation complexity, computational cost, temporal and spatial resolvability etc.) could be mentioned in the text.

Thank you for this suggestion. We have added the following sentence in the abstract and in the discussion: "We show that the RBF modelling technique can be used for processing GRACE data yielding global gravity field models which fit independent reference values at the same level as commonly accepted global geopotential models based on spherical harmonics."

We added (after "hydro-geophysical signals"):
Some key advantages of the method are summarized as follows:
- enhanced temporal resolution,
- regional solutions and refinements if local covariance information available,
- no post filtering required (user- friendly),
- spatial constraining (e.g. land/ocean de- coupling),
- linear equations and low computational costs,

- reduced artefacts through spatial localization compared to global coefficient estimation,
- combination with other space gravimetric techniques such as satellite laser ranging, gradiometry, and sea surface topography from altimetry.

2- The RBF-based modelling technique is described in Section 1 using the standard text with some references. I would like to see at least fundamental equations of the mathematical model to make the article self-contained for its readers.
The basic equations require several pages, and as they are given in (Gruber et al., reviewed in GJI) and (Novak, 2007) this would result in a third repetition.

3- The alternative global geopotential models are referenced in Section 1 but for the same reason given above, few words about each model could be added (maximum resolution, processing technique, estimation of low-degree coefficients etc.).

We have added at the end of manuscript:
For details about the maximum resolution, error estimates and low-degree harmonic coefficients the reader is refered to the corresponding file ftp://gfzop.gfz-potsdam.de/EGSIEM/readme.

---

## Author Response (AR2)

**Journal: ESurf**
Title: Earth's surface mass transport derived from GRACE, evaluated by GPS,
ICESat, hydrological modeling and altimetry satellite orbits
Author(s): Christian Gruber et al.
MS No.: esurf-2017-70
MS Type: Research article
Iteration: Major Revision

Dear Associate Editor,

thank you very much for your comments to the esurf-2017-70 manuscript. Please find below our answers (marked in blue) to your comments (given in black). We hope, the changes done in the revised manuscript and our response will allow to accept this manuscript.

With kind regards,

Christian Gruber (on behalf of co-authors)

**Associate Editor Decision: Reconsider after major revisions** (27 May 2018) by David Lundbek Egholm
Comments to the Author:
Based on the updated text and the authors' response to reviewers I suggest that the text is returned to the authors for major revision based on my comments. The text is in my opinion far from ready for publication. I will explain below:

• I fully agree with reviewer 1 that the methods used here need a thorough introduction, preferably in a separate methods section. The authors have largely ignored this reviewer suggestion because the methods are apparently described in a GJI paper under review. However, the GJI paper is not yet accessibly to Esurf readers, and, even if it was, the methods need more introduction here in order for this to a stand-alone-contribution. Please provide a methods section after the introduction in order to clarify to the readers how exactly the radial basis functions are used.

The RBF method used by us in the esurf-2017-70 manuscript is thoroughly described in the manuscript "Short latency monitoring of continental, ocean- and atmospheric mass variations using GRACE inter-satellite accelerations" still being under review at "Geophysical Journal International (GJI)". The revised version of this manuscript (attached) will be submitted to GJI next days. However, in order to mature the esurf-2017-70 manuscript into a self-consistent publication we have now added in this manuscript a separate section 'Methodology' describing the RBF-method used by us. Here, we refer also to the used Kernel function, derived by Novak (2007) that provides the basis for the transformation of the ranging observations into gradient-like values and their further mapping to the time-variable surface mass.

• The quality of the written English is still too poor, and it has not improved during revision. If improving the English is problematic for the author team they must seek professional help before resubmitting the text. I list a few examples below these general comments, but following up on those is not enough, as the text need thorough revision.

We have now significantly improved English in the whole manuscript. We consistently use present tense instead of past tense and perfect tense, when describing our analysis. Past tense is used when mentioning the research and results performed and obtained before this our study.

• The citation style is wrong in many cases, e.g. in lines 33, 51, 63, 129, 156.

We have checked the citation style and changed it according to the "Manuscript preparation guidelines for authors".

• The text is in often fragmented with paragraphs of one or two sentences (e.g. lines 144-146, 273-277, 326-328).

We agree with this comment of the Associate Editor and excluded the unnecessary fragmentation of the text at lines 158-160, 176-177, 182-183, 276-277, 291-292, 321-322, 363-364.

• Regarding structure, I would like to see sections 2-5 as subsections of a results section.

As we have mentioned in the response to the first comment, the detailed description of the RBF method is described in the GJI manuscript, while the Esurf manuscript is devoted to the validation of the RBF and a few other methods. Therefore we would prefer to keep the current structure of the manuscript.

line 24: Please add an extra s to 'asses'.

This has been done.

line 34: '… highly accurately' – please explain what you mean by this.

We have added the words "to only a few tens of µGal" to explain this.

line 43: '…gain further access to mass transport' do you mean something like 'in order to gain further knowledge/insight/information on mass transport…'

We have reformulated these words as "to gain further knowledge on mass transport".

line 51-55: This needs further explanation

As soon as the GJI manuscript Gruber et al. (in review) will be accepted, these words will be replaced by "Gruber et al., GJI, 2018, accepted".

line 63: '…has been applied by us…' In this study? If so, present tense is more appropriate. If not please give reference.

Yes, this is applied by us in this study. Therefore, we consistently use the present tense to describe our study and past tense when citing the studies performed before this our study.

lines 86-94: These bullet points were added in revision, but they are full of grammatical errors and typos.

This has been corrected.

line 112: doesn't -> does not

This has been corrected, as suggested by the Associate Editor.

lines 119-127: Why all the hyphens in this paragraph?

The hyphens have been erased in this paragraph in all cases, except the cases "center-of-figure", "center-of-mass" and "de-aliasing", since these are terms commonly used in scientific publications in this field of research.

line 141: 'thus' -> 'whereby' and 'was' → 'is'

The word "thus" has been replaced by "whereby".

line 142: 'is' -> 'are'

This has been corrected, as suggested.

lines 144-146: This looks to be a figure caption in the wrong place.

This sentence discussed Fig. 1. The unnecessary paragraph has been erased just after this sentence.

line 172: delete 'has'

Done.

line 176: delete 'thus'

Done.

line 223: 'it's' -> 'its'

This has been corrected.

line 230: 'find out' -> 'determine'

This has been replaced, as suggested.

line 233 and elsewhere: 'have been found' – the use of passive voice should be reduced.

This has been corrected here and also in many other cases.

line 245: 'Still it remains difficult'?

It has been changed to "Still, it remains difficult".

line 256: Please make sure to correct this one also

This sentence is, from our point of view, correct.

line 308: 'Their'?

replaced by
"The results of our evaluation confirm once again"

line 312: Please rephrase this too.

This sentence is, from our point of view, correct and clear.